# Prognostic Value of Pentraxin3 Protein Expression in Human Malignancies: A Systematic Review and Meta-Analysis

**DOI:** 10.3390/cancers16223754

**Published:** 2024-11-07

**Authors:** Hera Jung, Jeongwan Kang, Kang-Min Han, Hyunchul Kim

**Affiliations:** Department of Pathology, CHA Ilsan Medical Center, Goyang-si 10414, Gyeonggi-do, Republic of Koreakang210@chamc.co.kr (J.K.); kiekie53@hanmail.net (K.-M.H.)

**Keywords:** PTX3, malignancy, cancer, prognosis, biomarker, meta-analysis

## Abstract

Pentraxin 3 (PTX3) is a protein involved in controlling inflammation. Studies suggest that PTX3 plays both supportive and harmful roles in cancer development. To better understand its impact, we conducted a meta-analysis of the clinical data to assess how high levels of PTX3 are linked to cancer outcomes. We analyzed data extracted from the literature searched from databases including PubMed, Embase, Web of Science, MEDLINE, and the Cochrane library. Nine studies encompassing 1215 patients were included in the analysis. The results showed that elevated PTX3 expression is associated with poorer survival in cancer patients, with a pooled hazard ratio of 1.89. This connection was significant in both tumor tissue and blood serum. These findings suggest that PTX3 could be used as a prognostic marker for cancer and may serve as a potential target for therapies, particularly for patients at a higher risk of cancer-related deaths.

## 1. Introduction

Pentraxin (PTX) 3 is a protein that belongs to the PTX superfamily, which has diverse functional roles [1]. This superfamily is divided into two subfamilies based on the length of their N-terminal regions, short-chain PTX and long-chain PTX [2]. Prominent members of the short-chain PTX subfamily include C-reactive protein (CRP, also referred to as PTX1) and serum amyloid P (SAP, also known as PTX2) [3,4]. In contrast, the long-chain PTX subfamily comprises PTX3, PTX4, neuronal pentraxin 1 (NP1), and neuronal pentraxin 2 (NP2) [3,4]. The promoter region of the PTX3 gene contains binding sites for various transcription factors, including nuclear factor kappa light chain enhancer of activated B cells (NF-κB) and activator protein-1 (AP-1) [5,6]. The NF-κB binding site is implicated in transcriptional regulation by proinflammatory cytokines, whereas the AP-1 binding site is believed to regulate basal transcriptional activity [6]. Additionally, the PTX3 promoter contains binding sites for other transcription factors, such as Pu1, SP1, hypoxia-inducible factor 1 alpha (HIF-1α), CCAAT/enhancer-binding protein beta (C/EBPβ), and IL-6 (interleukin-6) [2,5,6]. PTX3 is synthesized by a variety of cell types, including inflammatory cells, mesenchymal cells, and epithelial cells [7]. It plays a crucial role in innate immunity and tissue repair by recognizing and binding to pathogens and cellular debris [8]. Consequently, PTX3 has been extensively studied in the context of inflammatory and infectious diseases [9,10,11].

There are conflicting reports regarding the role of PTX3 protein in cancer. The authors of some studies have demonstrated the tumor-suppressive effects of the PTX3 protein [12,13]. Conversely, PTX3 protein has also been reported to enhance tumor aggressiveness and reduce the host’s immune response against tumor cells [14]. Notably, high expression levels of PTX3 protein in serum and tumor tissues have been associated with poor prognoses in various human malignancies, including colorectal cancer, diffuse large B cell lymphoma, hepatocellular carcinoma, glioma, ovarian cancer, pancreatic cancer, primary myelofibrosis, and small cell lung cancer [15,16,17,18,19,20,21,22,23]. Consequently, PTX3 has been proposed as a potential pan-cancer prognostic marker [24].

In this meta-analysis, we aimed to comprehensively assess the prognostic value of PTX3 protein expression across a range of various human malignancies. As previously noted, PTX3 exhibits both protumoral and antitumoral effects, and there are only a limited number of studies on its prognostic significance in cancer. Therefore, through this analysis, we seek to elucidate the impact of PTX3 protein expression in human malignancies and clarify its potential role as a prognostic marker.

## 2. Materials and Methods

This study was registered on PROSPERO (CRD42024510311). The review and meta-analysis were conducted according to the PRISMA 2020 statement [25].

### 2.1. Study Design and Literature Search

A comprehensive literature search was performed on PubMed, Embase, Web of Science, MEDLINE, and the Cochrane Library on 20 January 2024. The search was conducted with the following combination of keywords: ((pentraxin 3) OR (pentraxin3) OR (pentraxin-3) OR PTX3 OR PTX-3) AND (cancer OR malignancy).

### 2.2. Inclusion Criteria for Literature Selection

Two authors independently reviewed the retrieved studies. The following criteria were applied: (1) studies reporting the prognostic significance of PTX3 protein expression; (2) studies utilizing immunohistochemistry or immunoassays for PTX3 protein detection; and (3) studies presenting results as overall survival (OS) using Kaplan–Meier curves.

### 2.3. Exclusion Criteria

The following exclusion criteria were applied: (1) articles unrelated to PTX3 expression; (2) non-original articles; (3) articles not published in English; (4) non-clinical studies; (5) studies with insufficient data; and (6) studies with low Newcastle–Ottawa scale (NOS) scores (5 or less).

### 2.4. Quality Assessment

Two investigators (H.J. and H.K.) assessed the quality of the studies included in the analysis with the Newcastle–Ottawa quality assessment scale. Disagreements were resolved through discussion.

### 2.5. Extraction of Data

Two investigators (H.J., and H.K.) independently extracted the following information from the studies: authors, year of publication, target molecule, and hazard ratio with standard error.

If the hazard ratio (HR) was not reported in the article, it was calculated using Engauge Digitizer software (version 12.1) following the method outlined by Irvine et al. [26]. When the HR, confidence interval, and p-value were provided in the article, they were used to calculate the standard error.

### 2.6. Statistical Analysis

R, version 4.3.2 [27], and the “meta” package, version 7.0-0 [28] were used for statistical analyses. To assess the prognostic significance of PTX3 protein expression, HRs and their standard errors were pooled. For these analyses, a random effects model was used. Given potential methodological differences among the studies, subgroup analyses were also conducted. Egger’s test, Begg’s test, and a funnel plot for HRs were performed in the analysis. Sensitivity analysis was performed using the leave-one-out method. *p*-values less than 0.05 were considered statistically significant.

## 3. Results

### 3.1. Characteristics of the Studies

The literature selection process is illustrated in Figure 1. The initial search yielded 2075 studies, from which 975 duplicate records were removed. An additional 976 studies were excluded based on their titles and abstracts due to their clear irrelevance. During the full-text review, we further excluded studies for the following reasons: 76 studies had insufficient data, 19 studies were non-clinical studies, 14 studies were non-original articles, and 6 studies were non-English articles. Ultimately, nine studies met the inclusion criteria for our meta-analysis.

**Figure 1 cancers-16-03754-f001:**
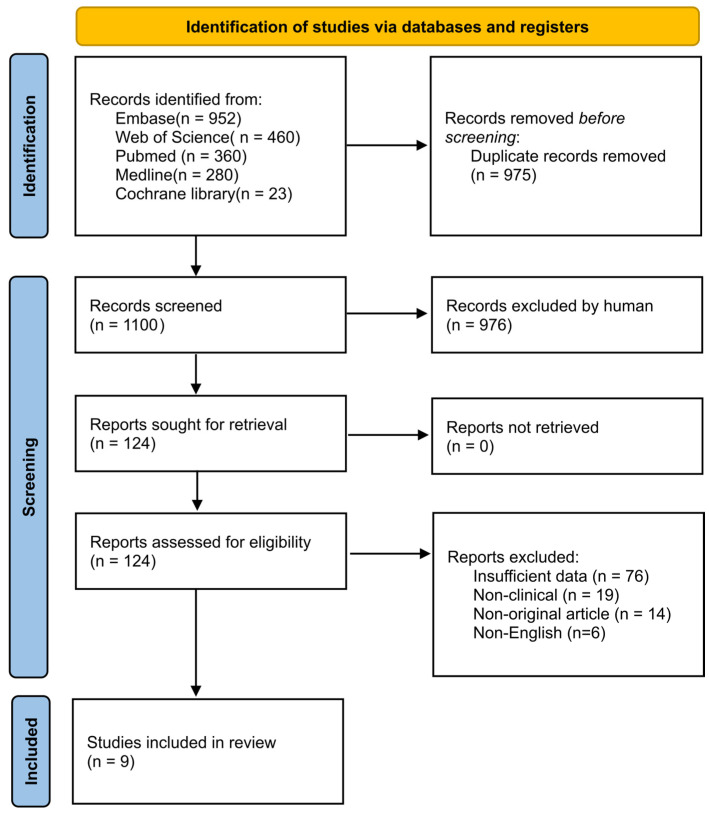
PRISMA flow diagram. The study populations were from China [16,17,19,20,21,23], Japan [15], and the United States [22]. The malignancies examined in these studies included colorectal cancer [19], diffuse large B cell lymphoma [15], hepatocellular carcinoma [17,21], glioma [23], ovarian cancer [16], pancreatic cancer [18], primary myelofibrosis [22], and small cell lung cancer [20]. Of the 9 studies, 5 used IHC method on tumor tissue [15,16,20,21,23] and 4 utilized immunoassay method on serum [17,18,19,22]. NOSs for the included studies were 7 or higher. The characteristics of the studies are summarized in Table 1.

### 3.2. Prognostic Significance of PTX3 Protein Expression

The HR for OS was pooled for PTX3 expression. There was a significant association between poorer OS (HR = 1.89, 95% CI = 1.55–2.32, *p* < 0.01) and high PTX3 levels with no significant heterogeneity (*I*^2^ = 0%, *p* = 0.89) (Figure 2).

### 3.3. Subgroup Analysis

A subgroup analysis was performed on the HR for OS in studies examining PTX3 expression. The data were categorized into two groups based on the study methods of IHC of tumor tissue and immunoassay of serum or plasma. The subgroup analysis revealed no heterogeneity within either group (IHC group: pooled HR = 1.93, *p* < 0.01, *I*^2^ = 0%, *p* = 0.7; immunoassay group: pooled HR = 1.86, *p* < 0.01, *I*^2^ = 0%, *p* < 0.01) (Figure 3).

### 3.4. Publication Bias

The tests for publication bias were conducted for HRs of OS for PTX3 expression. Statistically significant publication bias was detected when performing Egger’s test (*p* = 0.03) but not when performing Begg’s test (*p* = 0.21). The funnel plot is shown in Figure 4.

### 3.5. Sensitivity Analysis

Sensitivity analysis was performed using the leave-one-out method to evaluate the robustness of the analysis. The results show the stability of the pooled HR (Figure 5).

## 4. Discussion

In this study, we investigated the prognostic significance of PTX3 protein expression across various human malignancies. All of the studies included in this meta-analysis consistently demonstrated a worse prognosis in groups with higher PTX3 expression. Both serum and tumor tissue expression of PTX3 protein were associated with similar prognostic outcomes, with no observed heterogeneity. However, the possibility of bias in the studies selected for this analysis cannot be entirely excluded.

PTX3 plays essential roles in physiological processes, serving as a pattern recognition molecule for humoral innate immunity and as a key component in female fertility [7,29]. As a pattern recognition molecule, PTX3 recognizes and binds to various fungal, bacterial, and viral pathogens [7,29,30,31,32,33,34,35,36]. Upon binding, PTX3 facilitates pathogen elimination through phagocytosis and complement activation by interacting with the Fc gamma receptor or the complement system [7]. PTX3 also contributes to female fertility by supporting the reproductive system [37]. In the ovary, PTX3 is produced by cumulus cells, a type of granulosa cell surrounding the oocyte, and forms part of the cumulus matrix, an extracellular complex composed of tumor necrosis factor-inducible gene 6, hyaluronic acid, and inter-alpha-inhibitor [7]. The cumulus matrix aids in oocyte ovulation and provides protection to the oocyte [37].

PTX3 is known to play key roles in various pathological states, including infection, inflammation, and tissue damage [7]. When infection caused by fungus, bacteria, and viruses occurs, PTX3 functions as a protective molecule against these infectious agents [30,32,33,34,35,36]. This protective role of PTX3 is supported by studies involving PTX3-deficient mice, which showed higher susceptibility to fungal infections [38]. In inflammatory states, PTX3 regulates inflammation in two opposing ways, either through exacerbation or inhibition [7]. PTX3 can exacerbate inflammation by activating the complement system or recruiting inflammatory cells [39,40]. Evidence also suggests that PTX3 inhibits inflammation by interacting with Factor H, a regulatory molecule in the complement system, or by reducing the recruitment of inflammatory cells [41,42]. PTX3 also plays a critical role in tissue damage. It prevents excessive collagen buildup and platelet aggregation during tissue damage [7]. PTX3 interacts with fibrin and plasminogen to facilitate fibrinolysis, which aids in preventing collagen accumulation [10]. PTX3 also inhibits platelet aggregation by interacting with fibrinogen and collagen, exerting antithrombotic effects [43]. Lastly, PTX3 is implicated in tumor pathology.

PTX3 is recognized for its dual role in tumorigenesis, exhibiting both protumoral and antitumoral effects (Table 2). In an in vitro study involving hepatocellular carcinoma (HCC) cell lines, PTX3 was shown to facilitate tumor cell proliferation, growth, and epithelial–mesenchymal transition (EMT) [21]. In addition to promoting EMT, PTX3 may enhance other aggressive features of tumor cells, such as stemness [44]. In glioblastoma, PTX3 has been reported as promoting tumor progression by inhibiting autophagy [45]. This tumorigenic role has also been observed in in vivo experiments, in which the upregulation of PTX3 was found to promote the proliferation and metastasis of tumor cells [46,47]. Conversely, inhibition of PTX3 suppressed tumorigenicity and metastasis [48]. These protumoral effects of PTX3 have also been demonstrated in experiments with pancreatic, stomach, breast, cervical, high-grade brain, and prostate cancer cell lines. [18,48,49,50,51,52,53,54]. Despite these findings, there is also evidence supporting an antitumoral role for PTX3. In animal models, the inhibition of PTX3 was found to exacerbate oncogenic changes by suppressing protumoral inflammation through regulation of the complement cascade [12]. Additionally, in cell line experiments of bladder, breast, prostate, lung cancers, melanoma, and sarcomas, PTX3 functioned as an antitumoral factor. It decreased cell proliferation, motility, metabolism, stemness, and drug resistance by inhibiting fibroblast growth factor (FGF) and its receptor. [13,55,56,57,58,59].

Contrary to the proposed dual roles of PTX3 in tumorigenesis, all of the studies included in this meta-analysis consistently demonstrated worse prognosis in groups with high PTX3 expression. These findings align with the previous studies that have reported similar associations. While the studies included in this analysis focused on the relationship between high PTX3 protein expression and poor survival outcomes, there is also evidence linking PTX3 RNA expression with survival in cancer patients [45,61,62,63,64,65,66,67,68,69,70,71]. Studies on PTX3 RNA expression have involved the investigation of various malignancies, including gliomas [61,65], glioblastomas [45,66,68,70], HCC [62], head and neck squamous cell carcinoma [64,71], melanoma [69], papillary thyroid carcinoma [67], and lung cancer [63]. Although one study did not demonstrate statistically significant results [62], the majority reported a significant association between elevated PTX3 RNA expression and poorer survival outcomes [45,61,63,64,65,66,67,68,69,70,71]. In a comprehensive pan-cancer analysis, high PTX3 RNA expression was similarly associated with worse survival across various malignant tumors, leading the authors to propose PTX3 as a potential pan-cancer prognostic marker [24]. However, given that not all malignancies exhibit this association, further research into the role of PTX3 in tumorigenesis is necessary to accurately assess its value as a universal prognostic marker.

The expression levels of the PTX3 gene and protein appear to vary significantly across different tumor types and patient populations. This heterogeneity indicates that certain demographic groups may have a higher risk of cancer-related mortality. A study investigating single nucleotide polymorphisms (SNPs) in the PTX3 gene revealed that different PTX3 genotypes were associated with variations in tumor stage and metastatic status [72]. The researchers hypothesized that these polymorphisms might result in altered PTX3 protein levels [72]. Additionally, experimental studies on animals have demonstrated that environmental factors, such as cigarette smoking, can influence PTX3 expression in endothelial cells, suggesting that environmental factors may contribute to differential PTX3 expression in specific populations [73]. The relationship between PTX3 expression and obesity is also complex. Obesity has been linked to reduced plasma levels of PTX3 protein but concurrently increased PTX3 gene expression in adipose tissue [74]. In studies examining various types of cancer, higher tumor grades were generally associated with elevated PTX3 expression [13,17,48,65,75]. This differential expression may be governed by epigenetic mechanisms, including the regulation of upstream signaling pathways or hypermethylation of the PTX3 gene enhancer region [76,77]. These findings highlight the intricate regulatory networks influencing PTX3 expression and suggest that both genetic and environmental factors play critical roles in modulating its levels across diverse populations.

PTX3 is implicated in several key signaling pathways that may contribute to its protumoral effects. Research has shown that PTX3 plays a significant role in the PI3K/AKT, Toll-like receptor (TLR), c-Jun N-terminal kinase (JNK), NF-κB, Wnt/β-catenin, and signal transducer and activator of transcription 3 (STAT3) pathways [69,78,79,80,81,82,83,84,85,86,87,88]. The PI3K/AKT pathway, which is crucial for regulating cell proliferation, is often associated with oncogenesis, and PTX3 has been identified as a modulator of this pathway [78,82,85]. Similarly, the TLR superfamily, which primarily functions as an immune receptor, has been linked to cancer immunity, with PTX3 influencing the activity of specific TLRs [69,79,83]. The JNK family of kinases, which governs apoptosis and cell proliferation, is also involved in cancer progression, and its activation has been attributed to PTX3 [80,84]. Additionally, PTX3 has been shown to activate the NF-κB signaling pathway, a family of transcription factors that regulate DNA transcription and cell survival, which has known oncogenic potential [69,81,88]. The Wnt/β-catenin signaling pathway, well-recognized for its role in carcinogenesis, can also be triggered by PTX3 [81,86]. Lastly, STAT3, a transcription factor that regulates downstream signals for cell proliferation and apoptosis, has been found to promote tumor cell growth through PTX3 [87]. These findings suggest that PTX3 may contribute to cancer progression by modulating several oncogenic signaling pathways, highlighting its potential role in tumor biology.

As discussed above, PTX3 is linked to cancer immunity. Researchers speculate that it may help tumor cells evade the immune system [53]. The exact mechanism behind this immune escape remains unclear; however, several theories are proposed. One theory suggests that PTX3 overexpressed by tumors inhibits the complement system, potentially shielding tumor cells from immune detection [53] because, unlike in pathogen-bound states, fluid-phase PTX3 may block complement activation [89]. Another possible explanation is that PTX3 impedes antigen presentation by dendritic cells [90]. Additionally, PTX3-associated complement components may cause tumor cells to express molecules aiding immune evasion [60]. The results of a recent study suggest that PTX3’s protumoral effect could be due to immunity provided by PTX3-associated M2 macrophages [91]. However, there is an opposing argument that PTX3 suppresses protumoral complement system activation [12]. The authors of this opposing report demonstrated increased complement activation and a higher incidence of cancer development in PTX3 knockout mice [12]. These conflicting findings indicate that PTX3 might have both protumoral and antitumoral effects.

PTX3 plays a crucial role in regulating tumor cell migration and metastasis through various mechanisms. Inhibition of PTX3 has been shown to downregulate EMT-related molecules, such as vimentin and matrix metalloproteinase (MMP) 3, in head and neck squamous cell carcinoma cell lines [46]. Similarly, in cervical cancer cell lines, PTX3 knockdown resulted in reduced expression of key molecules involved in tumor cell migration and invasion, including MMP2, MMP9, and urokinase [48]. In melanoma cells, PTX3 production was found to stimulate the expression of EMT-related factors, such as TWIST1, further supporting its role in promoting tumor metastasis [69]. In breast cancer cells, PTX3 has been implicated in tumor cell migration through its regulation of protein kinase C ζ (PKCζ) activation [69]. Moreover, the results of studies on prostate cancer demonstrate a strong association between high PTX3 expression and tumor metastasis [72]. In glioma cells, PTX3 is suggested to be regulated by spen paralogue and orthologue C-terminal domain containing 1 (SPOCD1), further highlighting its involvement in metastatic processes [47]. These findings collectively indicate that PTX3 is a critical regulator within the complex signaling pathways that drive tumor metastasis, influencing key molecules and pathways responsible for cancer progression and dissemination.

The evidence suggests that PTX3 holds significant potential as a therapeutic target due to its pivotal role in various signaling pathways and tumor metastasis. Chivot et al. demonstrated that poly ADP-ribose polymerase (PARP) inhibitors downregulated tumor angiogenesis in breast cancer cells by inhibiting PTX3 expression [92]. In triple-negative breast cancer cells, PTX3 inhibition led to less aggressive tumor behavior through the TLR4 signaling pathway [79]. Similarly, the downregulation of PTX3 using an antitumor analgesic peptide in breast cancer cells reduced cancer cell stemness, EMT, migration, and invasion [81]. Further supporting these findings, the results of studies on neuroblastoma cells showed that PTX3 inhibition resulted in decreased stemness, proliferation, and migration [93]. In glioma cells, PTX3 knockdown was associated with a significant reduction in cell proliferation, migration, and invasion [54]. Additionally, in both in vitro and in vivo experiments involving cervical cancer cells, PTX3 gene knockdown led to decreased expression of proteins involved in cell cycle progression and migration, such as MMPs, and increased expression of cell cycle arrest proteins, thereby reducing the oncogenic and metastatic potential of the tumor cells [48]. These results underscore the therapeutic potential of targeting PTX3 to inhibit tumor progression, stemness, and metastatic behavior across various cancer types, suggesting that PTX3 inhibition could be an effective strategy in cancer treatment.

PTX3 appears to serve as a crucial link between inflammation and tumorigenesis, exerting protumoral effects through cancer-related inflammation. The regulatory role of PTX3 in inflammatory diseases is well established [4,7,8,10,11]. In pancreatic cancer, a positive correlation has been observed between high PTX3 expression and inflammatory markers, including serum CRP and IL-6 levels [18]. In experiments involving melanoma cells, PTX3 was found to interact with inflammatory signaling pathways, such as the TLR4 and NF-κB pathways [69]. Similarly, interactions between PTX3 and inflammatory pathways, including the IL-17 and tumor necrosis factor (TNF) pathways, were identified in a glioblastoma study [70]. Another study on glioblastoma demonstrated that PTX3 modulates the activity of tumor-infiltrating macrophages [23]. Similar findings were found in a gastric cancer cell study in which inhibition of the PTX3 gene suppressed cancer-associated inflammation through inhibition of the migration of macrophages [50]. In prostate cancer, PTX3 showed an association with the increased early component of the complement system and a complement inhibitor, CD59, and the authors speculated that the change in inflammatory signal possibly promoted the oncogenic process [60]. Similar changes in the level of the complement system were found in colorectal carcinoma patients [94]. The above evidence suggests that elevated PTX3 expression may promote protumoral inflammatory activity.

PTX3 appears to promote oncogenic inflammatory effects by modulating the tumor microenvironment, ultimately contributing to protumoral processes. Several studies on brain tumors have consistently shown an association between PTX3 expression and inflammation within the tumor microenvironment [23,66,68,95,96]. For instance, elevated expression of PTX3 and other inflammatory genes has been observed in brain tumor tissues compared to adjacent non-tumor tissues [95]. In cases of recurrent glioblastoma, PTX3 gene expression was found to be lower in the recurrent tumors than in primary tumors, indicating a possible dynamic interaction between PTX3 expression and the tumor microenvironment during tumor progression [68]. Further evidence supports the role of PTX3 in regulating the proliferation and migration of tumor-associated macrophages and dendritic cells in glioblastoma [23,66,96]. Similar associations between PTX3 expression and tumor-associated macrophages have also been observed in colon cancer [91]. In fibrosarcoma, tumor-infiltrating lymphocytes also appear to be influenced by PTX3 gene expression, suggesting its broader impact on immune cells within the tumor microenvironment [97]. In breast cancer, cancer-associated adipocytes have been shown to exhibit increased expression of PTX3, which plays a role in promoting more aggressive tumor behavior [98]. This finding was demonstrated in a co-culture experiment involving triple-negative breast cancer cells and adipocytes, where the upregulation of PTX3 expression led to enhanced aggressiveness of the cancer cells [99]. These findings suggest that PTX3 not only influences immune cells but also modulates other components of the tumor microenvironment, such as adipocytes, thereby contributing to cancer progression.

Several inflammatory biomarkers demonstrate diagnostic and prognostic value in various cancers, yet PTX3 has shown superiority over other inflammation-related biomarkers in cancer prognosis [100]. Commonly studied inflammatory markers such as CRP, IL-6, TNF-α, and procalcitonin (PCT) have all been associated with cancer-related inflammation [101] CRP, for instance, is recognized as a prognostic marker in multiple malignancies, including pancreatic, breast, gastrointestinal, and renal cancers [102,103]. Similar to PTX3, CRP contributes to tumor progression by modulating the tumor microenvironment, particularly through angiogenesis and inflammatory responses [102]. Elevated IL-6 expression has been correlated with poorer outcomes in cancer patients and is often linked to advanced tumor stages [104]. IL-6 is also a key player in paraneoplastic syndromes, further emphasizing its role in cancer progression [104]. Although findings on TNF-α are sometimes contradictory, the cytokine is generally associated with higher tumor stages and has potential as a prognostic marker [105,106]. PCT, primarily studied in the context of infectious diseases, has also shown promise as a prognostic indicator in cancer patients [107,108]. Remarkably, PTX3 has exhibited the highest sensitivity and specificity among biomarkers of cancer-associated inflammation [101]. In one study on colorectal cancer, PTX3 emerged as a significant prognostic factor; in comparison, other markers, such as IL-6 and TNF-α, did not demonstrate the same prognostic utility [109]. Unlike CRP, which is primarily produced in the liver, PTX3 is synthesized by cells within the tumor microenvironment, offering a more accurate reflection of tumor activity [53]. Additionally, PTX3 surpasses other members of the pentraxin family, such as NP1, NP2, and PTX4, in terms of its prognostic significance across a broad range of cancers [110]. Its widespread applicability further underscores PTX3’s potential as a superior biomarker for cancer prognosis.

PTX3 demonstrates considerable potential as both a diagnostic biomarker and therapeutic target in cancer. Studies have consistently shown elevated gene expression or serum levels of PTX3 in cancer patients, highlighting its diagnostic utility across various malignancies [16,111,112]. Higher PTX3 expression has been particularly associated with more aggressive cancer phenotypes, including increased cell migration, invasion, and metastasis [21,113]. Notably, inhibition of PTX3 has been found to reduce these aggressive characteristics, further emphasizing its role in cancer progression [113]. In addition to its correlation with cancer aggressiveness, PTX3 expression has been linked to the differentiation status of cancer cells [16]. Tumors exhibiting higher PTX3 expression are often poorly differentiated, suggesting that PTX3 may play a role in the maintenance of less differentiated, more malignant cell states. This association underscores the potential for PTX3-targeted therapies to be especially beneficial in treating aggressive, poorly differentiated tumors. Given these findings, PTX3 inhibition emerges as a promising therapeutic strategy, particularly for patients with cancers characterized by high PTX3 expression and aggressive behavior. Targeting PTX3 could provide a novel approach to mitigating tumor progression and improving patient outcomes, especially in cases where conventional therapies may be less effective against aggressive, poorly differentiated cancer cells.

This meta-analysis presents several novel aspects. Notably, it is the first meta-analysis to examine the prognostic significance of PTX3 protein expression in human malignancies. The focus on protein expression is particularly important because gene expression does not always correlate with the functional activity of the corresponding protein [114], and discrepancies between gene and protein expression levels are often observed [115]. Our analysis revealed a significant association between worse prognosis in malignant tumors and high PTX3 levels, both in tumor tissue and serum, suggesting that PTX3 could serve as a prognostic biomarker when measured in these biological samples.

However, this study also has several limitations. First, the total number of patients included in this analysis is relatively small, with 1215 patients across nine studies, which may affect the robustness of our conclusions. Second, all of the studies included in this analysis are retrospective case control studies, introducing the potential for selection bias. Third, the possibility of publication bias was detected in our bias detection test. The lack of studies with contrary findings could contribute to this bias, as indicated by the funnel plot, which shows an absence of data points in the lower left corner, representing low HR results. The absence of negative results can contribute to positive publication bias, potentially leading to an overrepresentation of favorable conclusions. To mitigate this issue, researchers and publishers should actively encourage the publication of studies with negative findings [116]. Future updated meta-analyses that incorporate additional studies on the prognostic significance of PTX3 expression will be crucial to reducing the impact of publication bias and providing more balanced and reliable results. The lack of negative results has also raised questions about the observed absence of heterogeneity in the findings. Zero heterogeneity is acceptable only if a sufficient number of studies are included and real-world variability can be deemed negligible [117]. However, as mentioned earlier, the relatively small number of studies included in this analysis limits the robustness of the results. Expanding the dataset with future studies will be essential to resolving the issue of heterogeneity and enhancing the reliability of the findings. Fourth, the HRs had to be extracted from Kaplan–Meier curves in most studies, as specific HRs were not reported. Although we utilized a recently developed, highly accurate method for this extraction, some degree of inaccuracy is inevitable. Lastly, we included only English language publications, which may have led to the exclusion of relevant data published in other languages.

## 5. Conclusions

In our meta-analysis, we found a significant association between high PTX3 levels and poor prognosis in patients with various malignancies, suggesting that the PTX3 protein could be used as a prognostic biomarker. Simultaneously, the results of this study suggest that PTX3 has potential as a therapeutic target. Given its higher expression in more aggressive malignancies, developing targeted therapies against PTX3 could benefit patients at a higher risk of poor outcomes. However, given the possibility of publication bias, we tentatively speculate that prognostic predictions might differ across different types of malignancies. To validate and further corroborate these findings, prospective studies with larger patient cohorts are required.

## Figures and Tables

**Figure 2 cancers-16-03754-f002:**
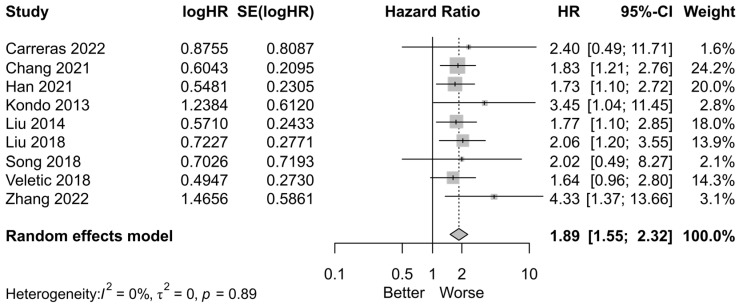
Forest plots for hazard ratios of overall survival in association with PTX3 expression (Carreras [15], Chang [16], Han [17], Kondo [18], Liu [19], Liu [20], Song [21], Veletic [22], and Zhang [23]). (HR: hazard ratio; SE: standard error; CI: confidence interval) (Black dot with gray box: point estimate, dotted line: confidence interval).

**Figure 3 cancers-16-03754-f003:**
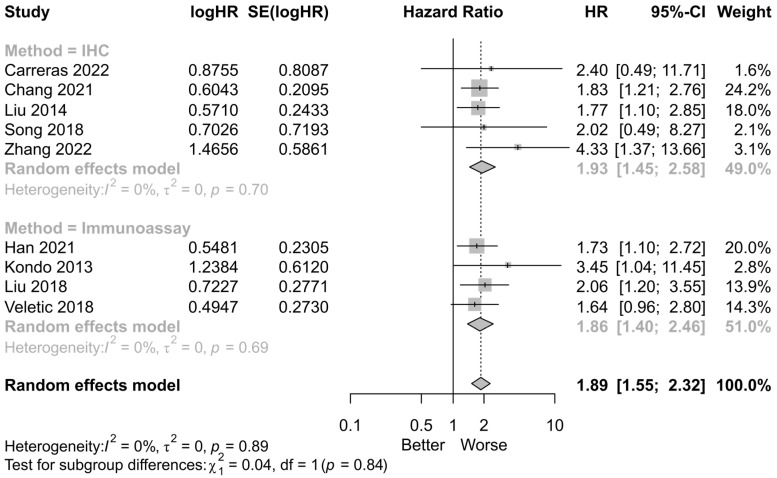
Forest plots for subgroup analysis by study methods (Carreras [15], Chang [16], Han [17], Kondo [18], Liu [19], Liu [20], Song [21], Veletic [22], and Zhang [23]). (HR: hazard ratio; SE: standard error; CI: confidence interval; IHC: immunohistochemistry) (Black dot with gray box: point estimate, dotted line: confidence interval).

**Figure 4 cancers-16-03754-f004:**
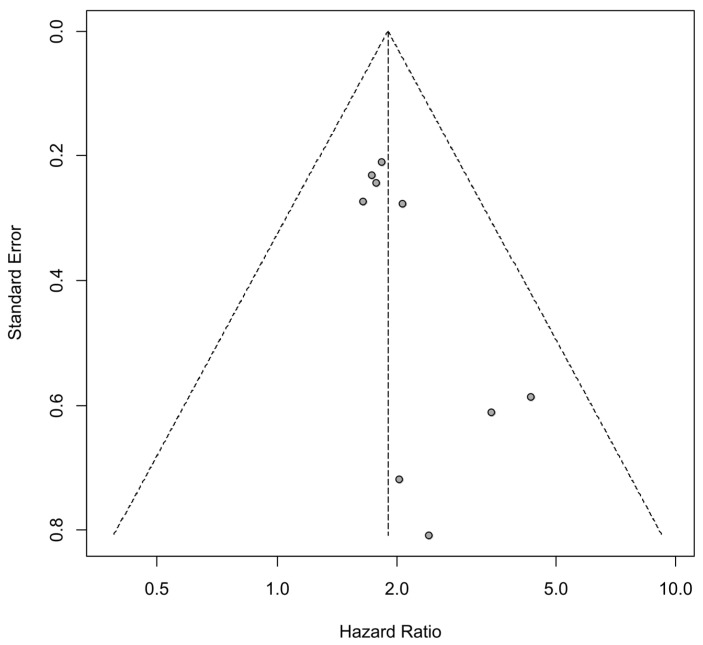
Funnel plot of hazard ratios of overall survival. (diagonal dotted lines: 95% confidence interval, vertical dotted line: overall effect, dots: individual studies).

**Figure 5 cancers-16-03754-f005:**
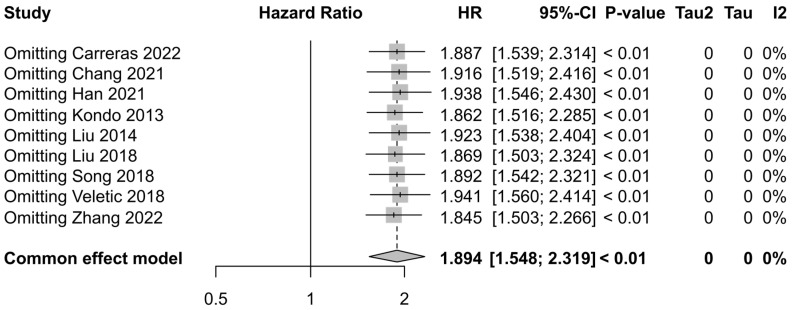
Funnel plot of sensitivity analysis with leave-one-out method (Carreras [15], Chang [16], Han [17], Kondo [18], Liu [19], Liu [20], Song [21], Veletic [22], and Zhang [23]). (HR: hazard ratio; SE: standard error; CI: confidence interval) (Black dot with gray box: point estimate, dotted line: confidence interval).

**Table 1 cancers-16-03754-t001:** Characteristics of the studies included in the analysis.

Studies	Country	Malignancy	Sample Size	Sample Type	Method	NOS
Carreras 2022 [15]	Japan	Diffuse large B cell lymphoma	148	Tumor tissue	IHC	8
Chang 2021 [16]	China	Ovarian cancer	168	Tumor tissue	IHC	7
Han 2021 [17]	China	Hepatocellular carcinoma	107	Serum	Immunoassay	7
Kondo 2013 [18]	Japan	Pancreatic cancer	78	Plasma	Immunoassay	8
Liu 2014 [20]	China	Small cell lung cancer	125	Tumor tissue	IHC	8
Liu 2018 [19]	China	Colorectal cancer	263	Plasma	Immunoassay	8
Song 2018 [21]	China	Hepatocellular carcinoma	158	Tumor tissue	IHC	8
Veletic 2018 [22]	U.S.	Primary myelofibrosis	140	Plasma	Immunoassay	7
Zhang 2022 [23]	China	Glioblastoma	28	Tumor tissue	IHC	7

**Table 2 cancers-16-03754-t002:** Protumoral and antitumoral effects and mechanisms of PTX3.

Protumoral Effects		
Tumor Type	Experiment Type	Mechanism
Hepatocellular carcinoma	Clinical and cell line study	Enhances proliferation, migration, invasion, and EMT [21]
Pancreatic cancer	Clinical and cell line study	Enhances migration [18]
Head and neck squamous cell carcinoma	Cell line and animal study	Enhances migration and invasion [49]
		Enhances metastasis [46]
Stomach cancer	Clinical and cell line study	Enhances migration [50]
	Clinical and cell line study	Enhances bone metastasis [51]
Breast cancer	Cell line study	Enhances bone metastasis [52]
	Cell line study	Promotes stemness and EMT [44]
Cervical cancer	Cell line and animal study	Enhances tumorigenesis and metastasis [48]
High-grade glioma	Cell line study	Promotes cell proliferation and invasion [54]
Glioma	Cell line study	Promotes cell proliferation and metastasis [47]
Glioblastoma	Cell line study	Modulates autophagy [45]
Prostate cancer	Clinical study	Recruitment of complement cascade [60]
Antitumoral effects		
Bladder cancer	Cell line study	Inhibition of FGF-driven proliferation and stemness [13]
Breast cancer	Cell line and animal study	Inhibition of DHT- and FGF8b-driven proliferation [55]
	Cell line and animal study	Inhibition of FGF2-driven proliferation and angiogenesis [56]
Melanoma	Cell line and animal study	Inhibition of FGF-driven proliferation and EMT [58,59]
Prostate cancer	Cell line and animal study	Inhibition of FGF-driven proliferation and angiogenesis [57,59]
Lung cancer	Cell line and animal study	Inhibition of FGF-driven proliferation and angiogenesis [59]
Sarcomas	Cell line and animal study	Regulation of tumor-promoting inflammation [12]

## Data Availability

The raw data supporting the conclusions of this article will be made available by the authors on request.

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
