# Peer review of "Prognostic Value of Pentraxin3 Protein Expression in Human Malignancies: A Systematic Review and Meta-Analysis"

_cancers, 2024, doi:10.3390/cancers16223754_

Round 1

Reviewer 1 Report

Comments and Suggestions for Authors

This systematic review and meta-analysis investigates the prognostic role of Pentraxin 3 (PTX3) protein expression in different cancers. The study synthesizes findings from nine studies, covering 1,215 patients, and shows that elevated PTX3 expression is associated with poorer overall survival (OS), supporting its potential as a pan-cancer prognostic biomarker. Both tumor tissue and serum levels of PTX3 were examined, showing consistent results with a pooled hazard ratio (HR) of 1.89. The authors deal with an up-to-date topic by exploring the role of PTX3 in cancer prognosis, thus providing a comprehensive meta-analysis that highlights its potential as a biomarker. The study is methodologically sound, following PRISMA guidelines, and the statistical analysis is robust. The manuscript is overall clear and well-written, I have only minor comments

  • As briefly mentioned in the Introduction and Conclusion sections, the point of PTX3 showing either protumoral and antitumoral effects in different tumors is one of interest, therefore the authors should show that in detail by adding a table
  • The authors are encouraged to discuss in more detail the potential impact of significant publication bias detected by Egger's test on the provided results of the meta-analysis. Are there ways to overcome this?
  • Heterogeneity in population, histotype, study design, is a big issue in meta-analyses, hence the authors should better discuss the reasons for the lack of significant heterogeneity in their analysis (lines 164-165)

Author Response

Comments 1: As briefly mentioned in the Introduction and Conclusion sections, the point of PTX3 showing either protumoral and antitumoral effects in different tumors is one of interest, therefore the authors should show that in detail by adding a table

Response 1: Thank you for the suggestion. I have added a table displaying the protumoral and antitumoral effects of PTX3 across different tumor types. (Line 210) Additionally, I have included more references in the paragraph discussing the dual effects of PTX3. (Line 200-209)

Comments 2: The authors are encouraged to discuss in more detail the potential impact of significant publication bias detected by Egger's test on the provided results of the meta-analysis. Are there ways to overcome this?

Response 2: Thank you for pointing this out. I have included an explanation of the potential impact of publication bias detected by Egger’s test and discussed possible ways to address the underlying causes of this bias. (Line 410-415)

Comments 3: Heterogeneity in population, histotype, study design, is a big issue in meta-analyses, hence the authors should better discuss the reasons for the lack of significant heterogeneity in their analysis (Line 164-165)

Response 3: Thank you for pointing this out. I have added a discussion on the lack of significant heterogeneity immediately following the sentences for response 2. (Line 415-421)

Reviewer 2 Report

Comments and Suggestions for Authors

The systematic review entitled “Prognostic value of pentraxin3 protein expression in human 2

malignancies: A systematic review and meta-analysis” by Jung et al., has a satisfactory content; however, major revision is required for publication in Cancers.

Major comments

-       I suggest highlighting the main characteristics of Pentraxin 3, in particular its biological function in physiological as well as in pathological conditions like cancer in the manuscript.

-       Pentraxin 3 plays a key role in the immune system. The authors should describe the involvement of immune system in cancer by focusing on Pentraxin 3. I suggest adding a paragraph focus on this to have a complete knowledge of Pentraxin 3. 

-       Future perspectives are missing in the paper. Please add it. It could be useful for the reader to better understand the perspectives of this study and the clinical impact. 

-       I suggest adding a table to summarize the abbreviations.

-       Please check carefully the manuscript to avoid typos. 

Comments on the Quality of English Language

Minor editing of English language is required.

Author Response

Comments 1: I suggest highlighting the main characteristics of Pentraxin 3, in particular its biological function in physiological as well as in pathological conditions like cancer in the manuscript.

Response 1: Thank you for pointing this out. I have added a detailed explanation of the biological functions of PTX3 in both physiological and pathological conditions. (Line 166-191) 

Comments 2: Pentraxin 3 plays a key role in the immune system. The authors should describe the involvement of immune system in cancer by focusing on Pentraxin 3. I suggest adding a paragraph focus on this to have a complete knowledge of Pentraxin 3.

Response 2: Thank you for pointing this out. I have included a more detailed explanation of the roles of PTX3 in the immune system. (Line 266-280)

Comments 3: Future perspectives are missing in the paper. Please add it. It could be useful for the reader to better understand the perspectives of this study and the clinical impact.

Response 3: Thank you for the suggestion. I have added a summary of the study's future perspectives and clinical implications in the conclusion, (Line 430-433) based on a paragraph from the discussion section. (Line 378-394)

Comments 4: I suggest adding a table to summarize the abbreviations.

Response 4: Thank you for the suggestion. I have added a table listing the abbreviations. (Line 437)

Comments 5: Please check carefully the manuscript to avoid typos.

Response 5: Thank you for the suggestion. I have carefully checked the manuscript to avoid typos.

Response to Comments on the Quality of English Language
Point 1: Minor editing of English language is required.

Response 1: The manuscript has been reviewed for English language quality through MDPI’s Author Service (Author Services ID: english-86584).

Round 2

Reviewer 2 Report

Comments and Suggestions for Authors

I recommend the publication of the manuscript.